# Experimental Study of the Damage and Failure Characteristics of the Backfill-Surrounding Rock Contact Zone

**DOI:** 10.3390/ma15196810

**Published:** 2022-09-30

**Authors:** Guang Li, Yang Wan, Jie Guo, Fengshan Ma, Haijun Zhao, Yanfang Wu

**Affiliations:** 1Key Laboratory of Shale Gas and Geoengineering, Institute of Geology and Geophysics, Chinese Academy of Sciences, Beijing 100029, China; 2Institutions of Earth Science, Chinese Academy of Sciences, Beijing 100029, China; 3College of Engineering and Technology, University of Chinese Academy of Sciences, Beijing 100049, China

**Keywords:** backfill mining, contact zone, Jinchuan mine, damage, failure, model test

## Abstract

Due to obvious differences in the properties of the filling body and surrounding rock, deformation always develops near the contact zone. Thus, determining the damage and failure characteristics of the contact zone between the backfill and surrounding rock is a precondition for safe production in mines. Taking Jinchuan mine as study area, the backfill-surrounding rock contact zones are divided into three models according to their different geometric shapes, namely, a linear model, embedded model, and multiple broken line model. A combined numerical simulation and physical model test method was adopted in this study. The research results show that the damage in the linear model begins at the seam, the failure is mainly concentrated in the filling body, and shear failure is dominant. The damage in the embedded model initially occurs around the inflection points, while the damage in the multiple broken line model initially occurs at the seams, and cracks always appear on the vertical contact surface first. Among the three contact models, the stability increases as follows: embedded > multiple broken line > linear. Moreover, the filling body enclosed by surrounding rock is the most stable, and the surrounding rock located in the footwall is more stable than the filling body located in the footwall. The conclusions of this study provide a theoretical basis for designing a mining scheme for Jinchuan mine and other mines with similar geological conditions and mining methods, and they provide a reference for studying the mechanical properties and stability of composite materials.

## 1. Introduction

The filling mining method can limit the deformation of the surrounding rock, control the changes in the ground pressure, and maintain the stability of the mine, which makes it an effective method that ensures safety in deep mining [1,2,3,4]. As a substitute for the ore body, backfill is used to form an interactive system that works together to resist external pressures from the overlying rock, ore body, and surrounding rock [5,6,7,8,9]. Since there is no sufficient deformation space, it is difficult for overall instability or extensive damage of the backfill body to occur, and sudden deformation failure mostly occurs in local parts of the backfill body, especially on the boundary [10,11,12]. Backfill instability accidents, surface cracking, shaft collapse, roadway destruction, and other disasters have revealed that the local stability of the backfill has a great impact on mine safety, and deformation failure of the backfill-surrounding rock contact zone (BSCZ) poses great hidden dangers in existing deep mine engineering [13,14,15,16]. For example, a sudden, violent roof caving event occurred in the Jinchuan nickel mine on 13 March 2016. The collapse involved an area of 11,000 m^2^, and the related surface subsidence involved an area of 19,000 m^2^, posing a considerable threat to the safety of both property and life [17]. Therefore, it is a crucial problem in backfill mining to determine the damage and failure laws of the BSCZ under different conditions.

Scholars have carried out a series of studies on the characteristics of the BSCZ through laboratory tests, theoretical analysis, and on-site monitoring [18,19,20]. As parts of the mining system, the backfill body and surrounding rock interact with each other to affect the mine stability. Therefore, Widisinghe et al. conducted their research from the point of view of the interaction mode between these two materials [21]. Aubertin et al. and Li et al. described the whole process of arch effect in filling body under the action of surrounding rock in detail [22,23]. By using a simplified mechanical model to solve complex mechanical problems, Liang et al. put forward a new analysis method of elastic modulus. Moreover, Ju et al. studied the mechanical properties of the backfill and surrounding rock and calculated the bending deformation of the roof [24,25]. After being filled with the underground goaf, the filling body experienced a phase transition, consolidation, and then provided a support force. Finally, the backfill body, the upper and lower orebodies and the surrounding rock formed a common system to resist external pressure [26,27]. The stability of the BSCZ is not determined by the strength of a single material, and thus, Nasir et al. studied the deformation characteristics of the contact surfaces according to the interface mechanics theory [28]. According to the test data, Rajeev et al. proposed a stress formula which was suitable for the contact surface between backfill and surrounding rock [29]. Liu et al. set equilateral triangular contact at the interface between backfill and surrounding rock, and the backfill stress distribution under different roughness interfaces was studied [30]. In addition, scholars often applied the research method of composite rock mass to the study of BSCZ. Li et al. established the creep model of the composite material and fitted the stress–strain curve [31]. Based on strain energy equivalence theory, Zhao et al. established the equivalent homogenization model of hard and soft rock mass, and the failure criterion of rock mass was derived [32]. Yu et al. obtained the fracture damage in surrounding rock and filling body by means of acoustic emission, and a method to describe the three-dimensional crack propagation was proposed [33].

The stability of the backfill is a mechanical problem related to the backfill, surrounding rock and the whole mining system, and the local instability has a serious impact on the global stability [34,35]. Although there has been considerable research on the BSCZ, studies on the local stability of backfill are relatively rare, especially the mechanical model and deformation mode of the BSCZ, and the deformation failure characteristics of the combined material of backfill and surrounding rock are not clear. Because of the different shapes of the boundary, there are various forms of contact between the filling body and surrounding rock. Different stress patterns lead to differences in the macro- and micro-characteristics of the damage evolution, deformation law, and failure mode of the BSCZ, as well as the damage formation mechanisms such as the crack initiation, triggering factors, and mechanical criteria. Therefore, it is necessary to further explore the damage and failure of the BSCZ.

In summary, Jinchuan mine, a typical backfilling mine, was taken as the study area. Based on a detailed field investigation, the damage and failure characteristics of the BSCZ were studied via numerical simulations and physical experiments. The goals of this study were to provide technical support for multi-stage filling mining in the Jinchuan No. 3.The expected research results can provide reference for the stability evaluation of other backfill mining mines with similar geological conditions.

## 2. Background

### 2.1. Engineering Geological Condition

Jinchuan mine, located in Jinchang City, Gansu Province, China, was discovered in 1958. The proven ore body was about 6500 m long and 500 m wide. The nickel metal reserves were about 5 million tons, ranking the second in the world; thus, the Jinchuan mine is of strategic significance [36]. The mining area has experienced multiple structural movements, including the Indosinian, Yanshan, and Himalayan movements, accompanied by strong magmatic intrusion. The geological structure is characterized by a simple fold shape and well-developed faults [37]. The mining area is arid and receives little rainfall, with an average annual rainfall of only 170 mm and an annual evaporation of 1700 mm [38]. The average surface elevation is about 1750 m, and the mine can be divided into four mining areas from west to east (Figure 1). Among them, the ore body in mining area No. 3 is inclined to the southwest, with a dip angle of 25–65° and a total length of 1300 m. The maximum thickness of the ore body is about 120 m, and the top pinch-out end of the ore body is about 200 m from the surface [39].

### 2.2. Mining Methods

Downward filling mining has been adopted in mining area No. 3 for several decades, resulting in the formation of a filling body with a huge volume in the goaf (Figure 2) [40]. The nickel grade (mass ratio) more than 1% is rich ore, less than 1% is poor ore. In order to further improve the mining efficiency, several working faces have been set up. The height of a single section is 20 m, and each section is divided into five sublevels with heights of 4 m. According to the actual mining schedule, the ore between +1210 m and +1190 m was excavated first via single section mining. Then, mining was begun at +1454 m, +1354 m, and +1254 m at the same time, and the sublevels with heights of 4 m were continuously mined downward until +1410 m, +1310 m, and +1210 m were reached, respectively. Finally, the remaining ore body was excavated through double section mining. The entire mining plan is shown in Figure 3.

It can be seen from Figure 2 that the geometric shapes of the BSCZs are complex and regular. The drift fill stoping method was employed in the study area, and the filling was immediately carried out after the mining, so the filling body had a regular rectangular boundary. According to the different geometric shapes, the BSCZs can be divided into 3 modes, including a linear model, embedded model, and multiple broken line model, which can be subdivided into five types from A to E (Figure 4).

### 2.3. Simulation of the Exploitation Process

Using the Fast Lagrangian Analysis of Continua 3-D (FLAC3D) software, the 36th line section was selected as the research object to simulate the entire resource mining process in the study area. The model had a size of 650 × 300 m, with elevations of 1750 m at the top and 1100 m at the bottom, and was divided into square grids with a side length of 4 m (Figure 5). The physical and mechanical parameters of the surrounding rock, ore body, and backfill adopted in the model are presented in Table 1. The horizontal displacement on the left and right sides and the vertical displacement on the bottom were limited [41]. Gradient in situ stresses were applied inside the model, and the ground stress in the study area was calculated as follows [42]:(1)$$\begin{array}{l} \sigma_{H}=1.083+0.034H\\ \sigma_{V}=0.028H-2.131 \end{array}$$
where *σ_H_* is the horizontal geostress (MPa); *σ_V_* is the vertical geostress (MPa); and *H* is the burial depth (m).

Figure 6a shows the distribution of the plastic zone around the mined-out area. It can be seen that the filling body and surrounding rock were relatively stable, and the plastic deformation mainly occurred near the contact zone, which was similar with the results obtained in the references [16,42]. Moreover, the failure was mainly concentrated in the filling body, and shear failure was dominant. In addition, the plastic zone was obviously concentrated near the thin filling body, where both the backfill and surrounding rock were prone to failure. According to the cloud diagram of the maximum shear strain increment shown in Figure 6b, it can be seen that strain concentration was most likely to occur at the inflection point of the contact zone, and dislocation deformation was most likely to occur on the contact surface. The five BSCZ types classified in Figure 4 exist in the actual mine, and strain concentration occurred around all of them, so they were weak positions affecting the global stability of the backfill. Excluding the increase in the shear strain caused by the thin layered backfill, BSCZs with similar burial depth were selected for comparative analysis. It was preliminarily concluded that for the multiple broken line model, the shear strain increment of type B, in which the backfill is located in the footwall, was greater than that of type C, and type C was more stable. In the embedded model, type E, in which the surrounding rock enclosed the filling body, was more stable than type D.

However, the damage and failure characteristics of the different BSCZ types could not be obtained from the numerical simulations described above, and further study was necessary.

## 3. Methodology

As an important research method, physical model tests have the advantages of concreteness, intuition, and reality, which play an important role in the study of complex engineering problems. In order to further explore the damage and failure characteristics of the BSCZs, physical model tests were used to reproduce the deformation processes of the five different contact types.

### 3.1. Simulation Relationship

As the theoretical basis of the mechanical model tests, the similarity ratio between the actual prototype and the test model was established. Dimensionality analysis is generally used to design the similarity ratio when the phenomenon is complex and the mechanism is not clear. Due to the limitations of the equipment, manufacturing process, and experimental materials, it is very difficult to achieve similarity in all aspects. Therefore, several major indicators can be selected according to the research goal to meet the similarity requirements.

Based on the Buckingham π theorem, the ratio of the same quantities in the prototype and model was defined as the similarity constant C. Through comprehensive consideration of the laboratory equipment and boundary conditions, the dimension of the physical model was set as 30 × 30 × 5 cm. Thus, the geometry similarity constant C_l_ was 40, and the other basic physical quantities were characterized in accordance with this ratio (Table 2) [43].

### 3.2. Simulation Materials

Selecting suitable test materials is the key to preparing rock-like materials. As rock is a heterogeneous and anisotropic material with a complex composition, cement, gypsum, and river sand were selected as the raw materials. Among them, river sand can increase the dilatancy, cement can increase the strength and fluidity, and gypsum can increase the brittleness and cementation. According to Table 1 and Table 2, the designed values of the physical quantities of the target material were calculated, and a series of similar material ratio tests were carried out with the sand–cement ratio and cement–gypsum ratio as variables. Finally, a river sand: cement ratio of 4:1 and a river sand: gypsum ratio of 4:1 were selected for the surrounding rock and backfill (Table 3) [16].

Self-designed polypropylene molds were used to make specimens of each BSCZ type. According to the needs of the test, the model frame with the size of 30 × 30 × 5 cm was fabricated, and the molds were designed according to the shapes of filling body and surrounding rock (Figure 7a). Firstly, the weights of cement, gypsum, and river sand were calculated based on the ratio of similar materials. Secondly, the ingredients were mixed well and water was slowly added. Then, the mixed material was poured into the frame which was painted with oil around the inner wall, and vibrated fully so that the bubbles could be discharged. A part of the specimen was fabricated first, and the other part was added after the first component was completely dry (Figure 7b–d). Finally, the models were demolded and the samples damaged in the process of maintenance and transportation were abandoned. Part of the specimens fabricated are shown in the Figure 7e.

### 3.3. Monitoring and Loading

During the experiment, the deformation failure processes of the BSCZs were recorded using a shooting system, including one digital camera for taking regular pictures and another camera for recording the entire process. Speckles were arranged on the front of each model to monitor the displacement and strain. A PCI-2 acoustic emission system produced by the American Physical Acoustics Corporation was used to collect the acoustic emission data. In order to ensure the speckle quality, acoustic emission sensors were attached to the back of the models. Acoustic emission monitoring points were installed on the surrounding rock, filling body, inflection point, and seam.

A hydraulic servo test platform was used to conduct the loading, and the maximum vertical and horizontal load could reach 300 kN. The operating system could control the horizontal and vertical loading separately, and the loading speed and time were set freely. The control system was controlled by the hydraulic servo in two mutually vertical directions of loading, and it could automatically adjust the loading speed and different two-way loading combinations. The displacements on the left and bottom boundaries of the model were limited, while the confining pressures were applied from the top and right boundaries, as shown in Figure 8. Considering the actual ground stress conditions and similarity relationship in the study area, the loading speed of the horizontal load was set as 0.04 kN/s, and that of the vertical load was set as 0.02 kN/s. The loading was applied until the sample was destroyed. The recording interval of the test data was 0.5 s [16].

## 4. Results

In order to highlight and compare the damage and failure characteristics of each type of BSCZ, the test results were divided into four parts: the images of the entire damage and failure process, the displacement field, the shear strain field, and the acoustic emission energy and energy accumulation curve.

### 4.1. Linear Model

Figure 9 shows the test results for BSCZ type A. As shown in Figure 9a, as the loading increased, a crack gradually developed along the diagonal direction of the filling body, and a shear failure state developed. The specimen was destroyed after the crack was transfixed and disconnected. As Figure 9b shows, the displacement field of the specimen was mainly in the horizontal direction, and the displacements in the vertical direction were small due to the support provided by the surrounding rock. In addition, the displacement was uniformly distributed in the surrounding rock, indicating that the overall integrity of the surrounding rock was good. In contrast, the distribution of the displacement in the filling body was uneven, especially near the joints, and the differential displacement was obvious. Figure 9c shows that the strain concentration of the specimen mainly occurred around the cracks, and the tension shear and pressure shear appeared alternately. It can be seen from Figure 9d that the damage to the specimen occurred at the beginning of the loading, and then it entered a plateau phase. Several strong energy release events occurred in the later stage, which led to failure, and the progression from crack penetration to final failure occurred in a very short period of time. Overall, the failure of the filling body led to instability failure of specimen type A. In other words, the backfill strength determined the stability of the BSCZ for the linear model.

### 4.2. Embedded Model

Figure 10 shows the test results for BSCZ type B. As shown in Figure 10a, as the loading increased, the crack formed in the filling body near the inflection point below the embedded surrounding rock first, and the crack expanded diagonally to the backfill. Finally, the filling body broke along the longitudinal seam. As shown in Figure 10b, the displacement field of the specimen diverged, with the surrounding rock as the center, and the deformation increased with increasing distance from the surrounding. Figure 10c shows that the strain concentration in the specimen mainly occurred around the cracks, the longitudinal penetrating cracks that caused the failure of the specimen developed at the joints between the two materials, and the tension–shear strain concentration occurred at these joints. It can be seen from Figure 10d that the specimen was relatively stable and did not produce significant energy release at the beginning of the loading. In the range of 130 s to 180 s, three concentrated energy release events occurred successively, which led to the final failure.

Figure 11 shows the test results for BSCZ type C. As shown in Figure 11a, as the loading increased, the fractures also appeared around the inflection point between the two materials at the beginning. However, in contrast to type B, the fractures did not develop toward the surrounding rock vertically, but they extended in two mutually distinct vertical directions along the seam until the cracks were connected. They finally exhibited inverted Y-shaped failure in the filling body. As shown in Figure 11b, the displacement of the specimen was mainly horizontal displacement, and the displacement of the surrounding rock on the right side of the filling body was large, which was the result of the overall inward extrusion of the surrounding rock. Figure 11c shows that an X-shaped strain concentration zone was formed at the inflection point in the upper right part of the filling body. The strain along the horizontal joint was mainly tension–shear strain, while the strain along the vertical joint was alternating tension–shear strain and compression–shear strain. It can be seen from Figure 11d that several energy release events occurred during the loading process for a long time, and these events corresponded to the generation and development of microscopic cracks in the model. When it was loaded to about 1000 s, the specimen experienced strong energy release and finally failed.

By comparing the two types of embedded contact models, it was found that the failure of the embedded model occurred at the inflection point between the two materials first, and then the cracks expanded along the seams until they were connected. Therefore, in these types of BSCZs, the inflection point strength was the weakest, followed by the seam. Moreover, due to the high proportion of surrounding rock, the stability of type C was higher than that of type B.

### 4.3. Multiple Broken Line Model

Figure 12 shows the test results for BSCZ type D. As shown in Figure 12a, in the initial stage of loading, longitudinal cracks formed at the seam between the filling body and surrounding rock. When the cracks reached the inflection point, they began to expand into the filling body. Two mutually distinct perpendicular cracks formed in the filling body and continued to develop. Finally, the cracks penetrated through the entire model, leading to the final failure. As shown in Figure 12b, the displacement of the specimen was mainly horizontal, the entire displacement field was oriented toward the filling body, and a large amount of displacement occurred around the seams. Figure 12c shows that a strain concentration zone running through the entire model formed inside the filling body, and it exhibited strong tensile shear strain. It can be seen from Figure 12d that the loading time of the specimen was short, and energy release events constantly occurred during the loading process. In addition, the energy released by each event was small, which was related to the large proportion of backfill, and the model exhibited strong plastic characteristics.

Figure 13 shows the test results for BSCZ type E. As shown in Figure 13a, in the initial stage of loading, cracks formed at the vertical seam between the filling body and the surrounding rock and then propagated into the filling body until they were connected. The failure mainly occurred inside the backfill. As shown in Figure 13b, the displacement field of the specimen pointed to the lower left of the model, and the horizontal and vertical displacements were similar. A displacement transition zone formed near the joint between the filling body and surrounding rock. Figure 13c shows that an arc-shaped strain concentration zone was formed around the seam, exhibiting strong compression–shear characteristic. It can be seen from Figure 13d that the energy release from the specimen was relatively stable during the loading process, and the cumulative energy curve increased nearly linearly. When the model was loaded for 130 s, a series of high-energy release events occurred, with a large amount of energy released in each event, indicating that serious sudden failure occurred in the model.

By comparing the two types of multiple broken line models, it was found that the failure occurred at the seams between the two materials first, and the cracks always formed in the vertical contact surface first. Then, the cracks spread along the seams until they coalesced, and the failure mainly occurred inside the filling body. Moreover, due to the high proportion of surrounding rock, the stability of type E was better than that of type B.

## 5. Discussion

### 5.1. Strength Comparison of Different Contact Zone Types

The horizontal stress–strain curves of each type of BSCZ are shown in Figure 14. It can be seen from Figure 14 that the strengths of the different types of BSCZs from largest to smallest are C, B, E, D, and A. That is, the strength of the embedded model was greater than that of the multiple broken line model, and the linear model was the most unstable. This is in good agreement with the results of the numerical simulations and the phenomena observed in the model tests.

In the linear model, there was a natural penetrating contact surface between the different materials, which was very detrimental to the stability. In the embedded model, the strength of the type in which the filling body was embedded in the surrounding rock (type C) was greater than that of the type in which the backfill was embedded in the surrounding rock (type B). This occurred because as the outer shell, the hard surrounding rock bore the energy induced by the load, and the deformation was slowly transferred to the internal filling body, so the overall stability was maintained. In addition, it can be seen from the stress–strain curve that types B and C almost lost all their bearing capacity after the stress reached the peak. If this damage occurred in practical production, the overall stability of the project would be greatly affected. For the multiple broken line model, because the ore body in the study area has a certain dip angle, in order to improve the mining efficiency, the multiple broken line model of the BSCZ is the most common in the engineering scheme. Type D always occurred in the footwall of the ore body, and type E often occurred in the footwall. The results reveal that type E is more stable than type D, and thus, more attention should be paid to the stability of BSCZs in the hanging wall.

### 5.2. Analysis of Acoustic Emission Characteristics

Figure 15a shows the cumulative energy curve and cumulative count curve of the acoustic emissions from the different types of BSCZs. It can be seen from Figure 15a that the cumulative energy released from each specimen was almost positively correlated with the specimen’s strength; and the greater the strength was, the more energy was released. The energy released from type C was much higher than those released from the other specimens, which may be related to the longer duration of the test. The shape of the cumulative count curve is almost the same as that of the cumulative energy curve. However, in type D, more counts were used to release less energy than in type E, which may be related to the different proportions of the filling body and surrounding rock in each BSCZ.

Figure 15b shows a comparison diagram of the acoustic emission energy and counts at different positions in the BSCZ. It can be seen from Figure 15b that the acoustic emission energy and counts from the surrounding rock occupied a small proportion, while the acoustic emission activities in the backfill, seam, and inflection point were similar, which is consistent with the observations made during the tests; that is, the damage initiated in the seams or inflection points and gradually propagated into the filling body, indicating that these positions were unstable and were prone to failure. It should be noted that the type A specimen had no inflection point, so there was one less sensor at the inflection point than at the other positions. In addition, the surrounding rock released more energy with a smaller acoustic emission count ratio, indicating that the single energy release capacity of the surrounding rock was stronger.

## 6. Conclusions

A classification method of BSCZ was proposed. Based on the shape of the filling body in the study area, the BSCZs can be divided into three models according to the geometric shapes: a linear model, embedded model, and multiple broken line model. Among them, the embedded model includes a type in which the filling body is wrapped by the surrounding rock and a type in which the surrounding rock is wrapped by the filling body, while the multiple broken line model includes surrounding rock located in the hanging wall and surrounding rock located in the footwall. The clear classification simplifies the complex contact zone and is beneficial to follow-up research.The damage of the linear model began in the seam, the failure was mainly concentrated in the filling body, and shear failure was dominant. The damage of the embedded model occurred at the inflection point, and the cracks propagated along the seams until they were connected. The damage of the multiple broken line model initially occurred in the seams, and the cracks propagated along the seams until they coalesced. The inflection points and seams were the most unstable positions, stress concentration easily developed around the inflection point, and shear deformation easily occurred in the seams.The embedded contact zone had the strongest strength. The type in which the filling body was enclosed by the surrounding rock was stronger than the type in which the surrounding rock was enclosed by the filling body. The strength of the multiple broken line model was the next strongest. The type in which the surrounding rock was located in the footwall was more stable than of the type in which the filling body was located in the footwall. Moreover, the stronger type was often located in the footwall, so more attention should be paid to the stability of BSCZs in the hanging wall.In the design of the mining approach, attention should be paid to avoiding a linear model contact zone, and the occurrence of thin layered filling should also be avoided. Although the embedded model is locally stable, the embedded part should be as short as possible. In actual production and monitoring, more attention should be paid to the inflection points in the embedded model and the seams in the multiple broken line model. In addition, support should be provided when necessary; for example, anchor bolts should be installed perpendicular to the joint to increase the friction at the joint and optimize the self-bearing capacity of the contact zone.
The damage to and deformation of the BSCZ is a regional, complex, and nonlinear system problem. There are many limitations of this study. The physical model tests and numerical simulation calculations both simplified the problem to a two-dimensional planar problem, which fails to reflect the complex engineering situation. In the physical model tests, the radial displacement of the model was not limited, which resulted in distortion of the speckle monitoring. In the process of loading the model, the vibration generated by the oil pump also affected the ability of the camera system to accurately capture the deformation, which should be improved in future research. Moreover, on the basis of this paper, the boundary types of BSCZ can be divided into more types. Furthermore, the experimental scheme can be optimized to analyze the deformation and failure mechanism of the BSCZ from multiple perspectives.

## Figures and Tables

**Figure 1 materials-15-06810-f001:**
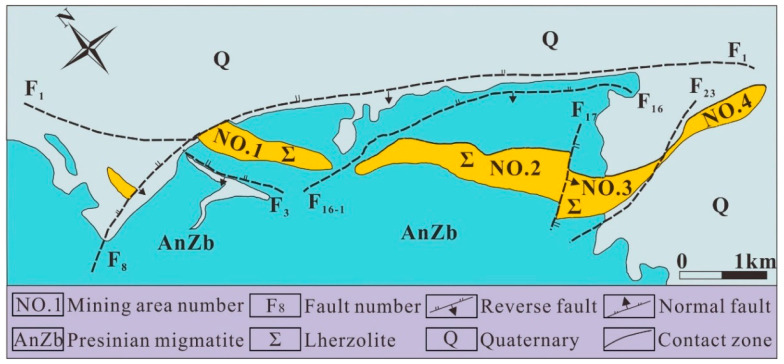
Engineering geologic map of Jinchuan mine [39].

**Figure 2 materials-15-06810-f002:**
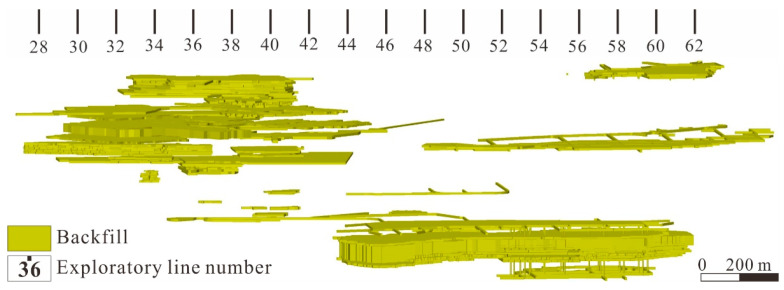
Huge volume filling body in the study area [40].

**Figure 3 materials-15-06810-f003:**
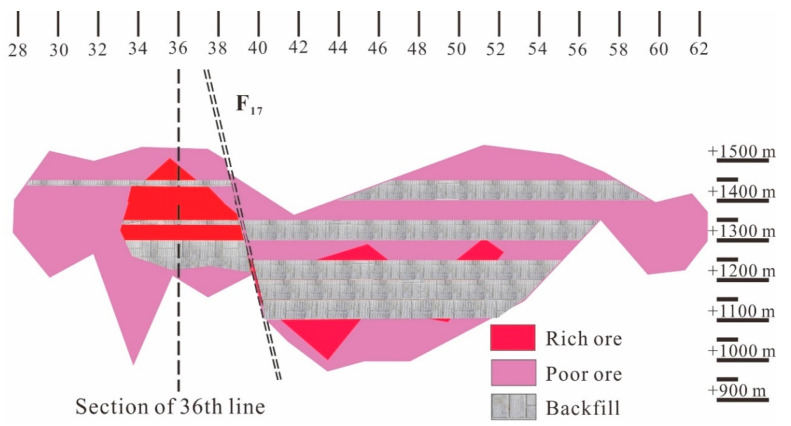
Cutaway schematic drawing of mining area No. 3.

**Figure 4 materials-15-06810-f004:**
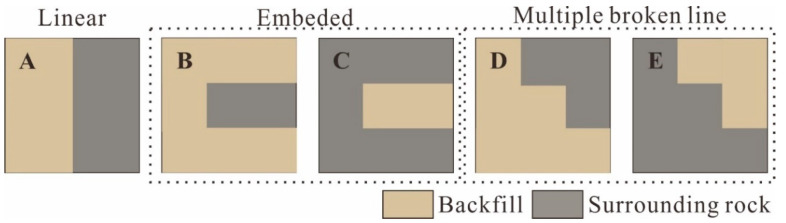
Classification of backfill-surrounding rock contact zones. (**A**) Linear model; (**B**) Embedded model with the filling body enclosed by surrounding rock; (**C**) Embedded model with the surrounding rock enclosed by filling body; (**D**) Multiple broken line model with the backfill located in the footwall; (**E**) Multiple broken line model with the surrounding rock located in the footwall.

**Figure 5 materials-15-06810-f005:**
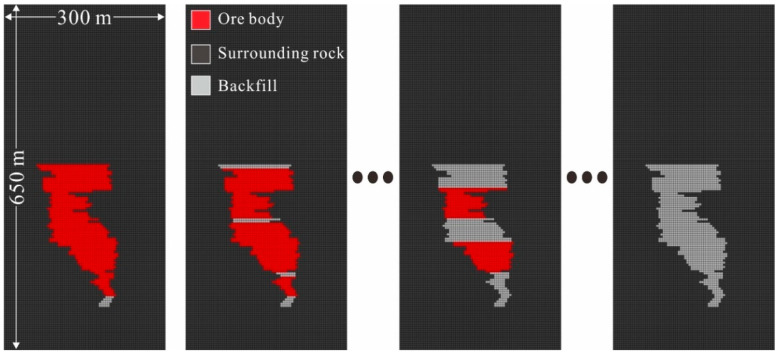
Numerical model and mining-filling steps.

**Figure 6 materials-15-06810-f006:**
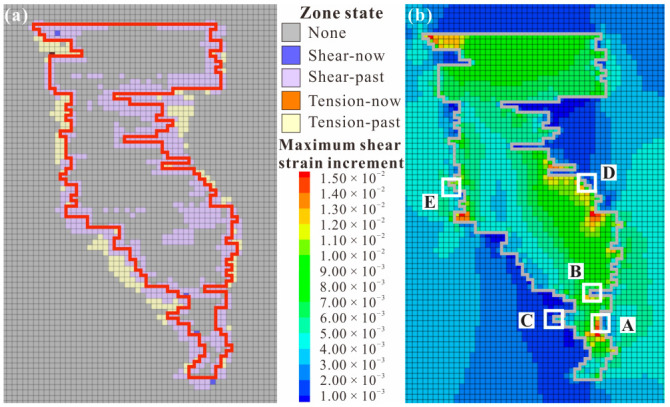
Numerical simulation results. (**a**) Plastic zone; (**b**) maximum shear strain increment. (A) Linear model; (B) Embedded model with the filling body enclosed by surrounding rock; (C) Embedded model with the surrounding rock enclosed by filling body; (D) Multiple broken line model with the backfill located in the footwall; (E) Multiple broken line model with the surrounding rock located in the footwall.

**Figure 7 materials-15-06810-f007:**
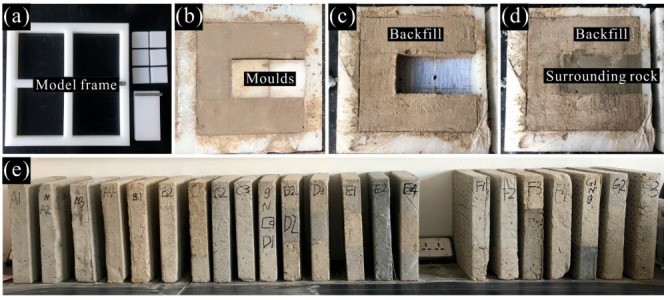
Physical model samples. (**a**) Model frame; (**b**) molds; (**c**) strain field; (**d**) acoustic emission energy; (**e**) acoustic emission energy.

**Figure 8 materials-15-06810-f008:**
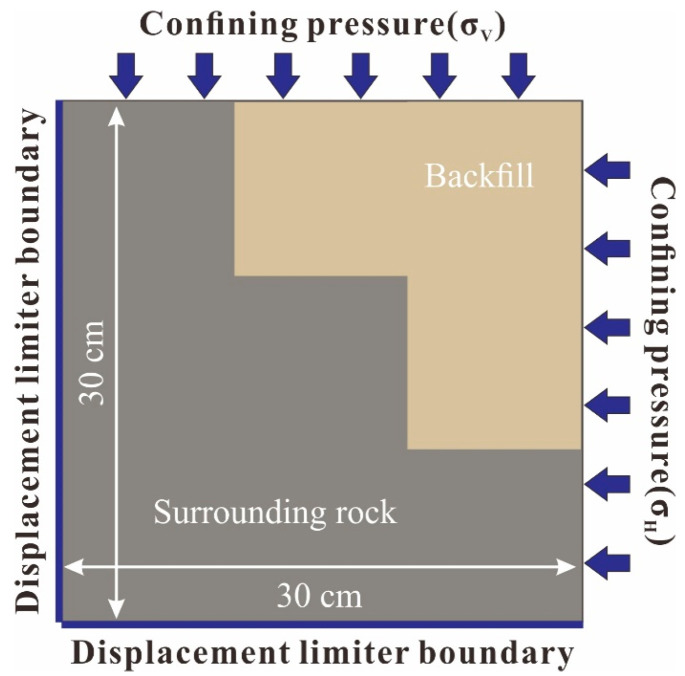
The boundary conditions of the physical model samples.

**Figure 9 materials-15-06810-f009:**
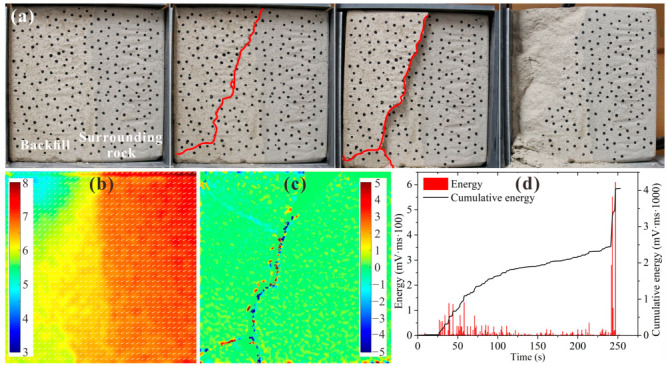
Experimental results for type A. (**a**) Damage and failure processes; (**b**) displacement field; (**c**) strain field; (**d**) acoustic emission energy.

**Figure 10 materials-15-06810-f010:**
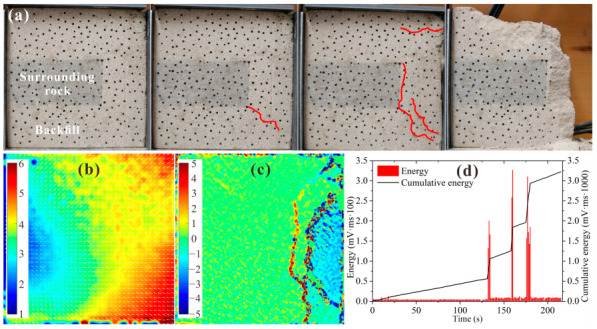
Experimental results for type B. (**a**) Damage and failure processes; (**b**) displacement field; (**c**) strain field; (**d**) acoustic emission energy.

**Figure 11 materials-15-06810-f011:**
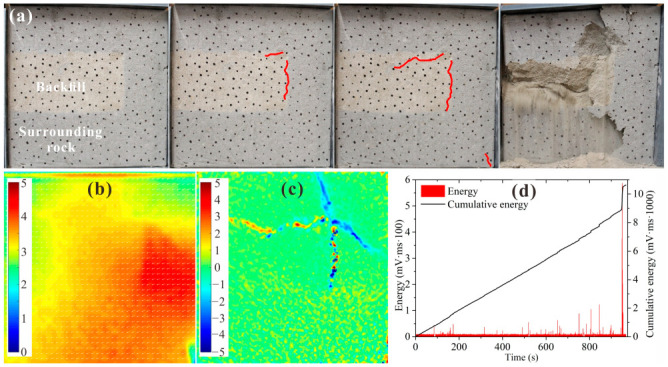
Experimental results for type C. (**a**) Damage and failure processes; (**b**) displacement field; (**c**) strain field; (**d**) acoustic emission energy.

**Figure 12 materials-15-06810-f012:**
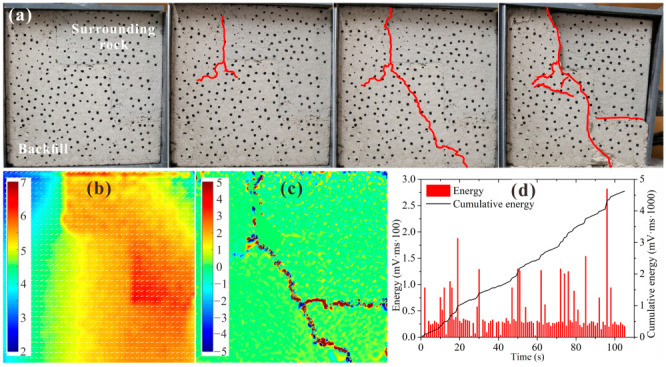
Experimental results for type D. (**a**) Damage and failure processes; (**b**) displacement field; (**c**) strain field; (**d**) acoustic emission energy.

**Figure 13 materials-15-06810-f013:**
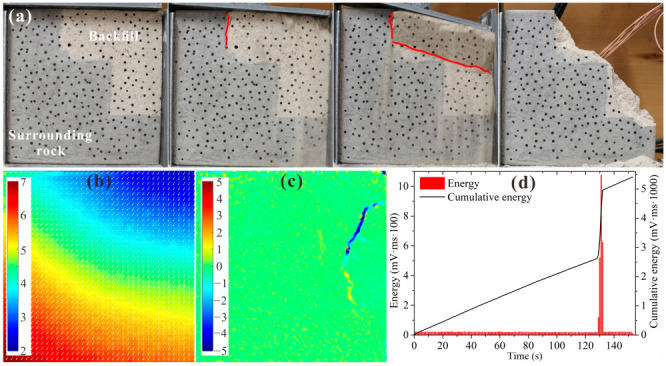
Experimental results for type E. (**a**) Damage and failure processes; (**b**) displacement field; (**c**) strain field; (**d**) acoustic emission energy.

**Figure 14 materials-15-06810-f014:**
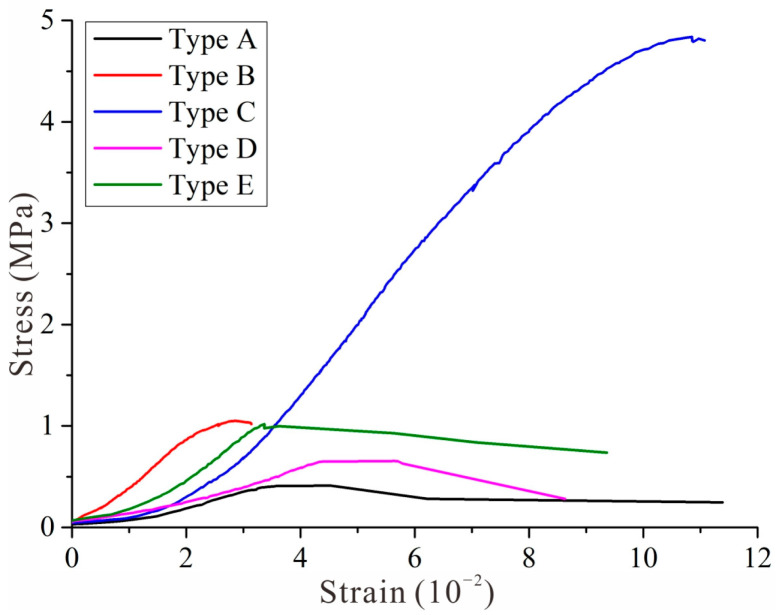
Stress–strain curves.

**Figure 15 materials-15-06810-f015:**
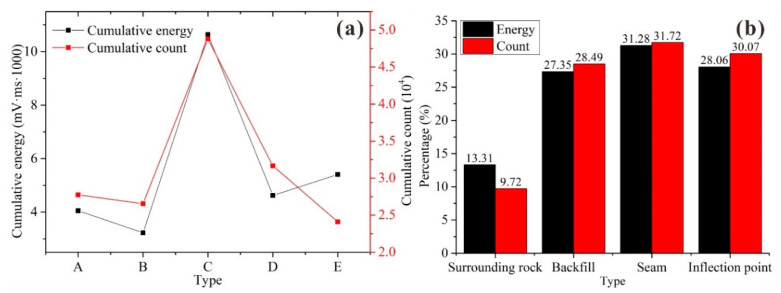
Acoustic emission data statistics. (**a**) Energy and cumulative energy curves; (**b**) energy and count statistics.

**Table 1 materials-15-06810-t001:** Rock and backfill mechanical parameters in mining area No. 3 [42].

Type	Density (g·cm^−3^)	Tensile Strength (MPa)	Compressive Strength (MPa)	Cohesion (MPa)	Internal Friction Angle (°)	Elastic Moduli (GPa)	Poisson’s Ratio
Surrounding rock	2.6	12.2	152	13.5	35	64	0.28
Ore body	3.05	0.5	16.4	4.6	41.7	8.6	0.31
Backfill	2	0.800	9.900	0.95	38	7.28	0.32

**Table 2 materials-15-06810-t002:** Similarity ratio.

Physical Quantity	Similarity Relation	Similarity Constant
Geometry (key constant)	C_l_	40
Density (key constant)	C_ρ_	1.25
Displacement (key constant)	C_D_	40
Poisson’s ratio	C_μ_ = 1	1
Elasticity modulus	C_E_ = C_ρ_C_l_	50
Strain	C_ε_ = C_ρ_C_l_/C_E_	1
Stress	C_σ_ = C_l_C_γ_	50
Internal friction angle	C_φ_ = 1	1
Cohesion	C_c_ = C_ρ_C_l_	50

**Table 3 materials-15-06810-t003:** Rock and backfill mechanical parameters in mining area No. 3 [16].

Lithology	Type	Density (g·cm^−3^)	Tensile Strength (MPa)	Compressive Strength (MPa)	Cohesion (MPa)	Internal Friction Angle (°)	Elastic Moduli (GPa)	Poisson’s Ratio
Surrounding rock	Design value	2.08	0.244	3.04	0.27	35	1.28	0.28
	Similar material	1.860	0.226	3.01	0.292	28.723	0.787	0.267
Backfill	Design value	1.682	0.051	0.392	0.176	22.079	0.192	0.199
	Similar material	1.6	0.016	0.198	0.019	38	0.146	0.32

## Data Availability

Not applicable.

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
