# Peer review of "Experimental Study of the Damage and Failure Characteristics of the Backfill-Surrounding Rock Contact Zone"

_materials, 2022, doi:10.3390/ma15196810_

Round 1
Reviewer 1 Report
Review report: Experimental study of the damage and failure characteristics of the backfill-surrounding rock contact zone
1. Omit the unnecessary information from the abstract section and add only key information.
2. Discuss the Novelty and clear application of the work.
3. Introduction section looks very week. Add key published work and try to make a bridge between current and previous published work.
4. Please check the quality of Fig. 2 and also add the reference for each one.
5. How were the mechanical properties evaluated in Table 1?
6. Add the reference for each equation.
7. Add the boundary condition and complete detail about the modelling part.
8. Fracture surface needs depth study along with proper mechanism.
9. Discussion section need depth study and should be focused on current work.
10. Shorten the length of eth conclusion section. Add only key bullet points.
11. In present it looks like a technical report not a paper. Revise the manuscript carefully.
Reviewer 2 Report
The paper is well conceived with only minor spelling errors (e.g. infection).
It is a valuable contrinution.
Reviewer 3 Report
Reviewer Comments
Paper title: Experimental study of the damage and failure characteristics of 2 the backfill-surrounding rock contact zone
The present manuscript describes a combined numerical simulation and physical model test method. The research results show that the damage in the linear model begins at the seam, the failure is mainly concentrated in the filling body, and shear failure is dominant. The damage in the embedded model initially occurs around the inflection points, while the damage in the multiple–broken line model initially occurs at the seams, and cracks always appear on the vertical contact surface first.
A manuscript has a practical application and also provides important theoretical for the next studies.
The paper can be accepted for publication after providing the corrections mentioned below.
Point 1. Keywords need to be modified. Please use words not combinations of words or phrases.
Point 3. In the Introduction section, an enhanced literature review is required. For this study, the authors have used only 23 reference sources. It seems insufficient for such type of research. It will be great if the authors show some description in context – Why it is important to conduct this study?
Paint 4. Can the expected result be used or implemented within other geological conditions? If yes, then how? What limitations?
Point 4. The aim and the tasks must be highlighted at the end of the Introduction section.
Point 5. Figure 2 is unclear. Please add more details in the description of the legend.
Point 6. You should add what does mean rich and poor ore. What is the classification.
Point 7. Section 2.1. What does the Project description mean?
Point 8. Boundary conditions should be given in more details.
Point 13. The novelty of the paper must be highlighted in the conclusions section.
Point 13. Please provide a short description of further research.
Point 14. Comparison with the previously achieved results are welcome.
Point 15. There are papers that I have reviewed in the past years. Please consider the suggested research in your paper when enhancing the literature review (Be aware that there are no references that belong to the reviewer). I believe they are worth considering in your paper.
1. Iordanov, I., Novikova, Yu., Simonova, Yu., Yefremov, O., Podkopayev, Ye., & Korol, A. (2020). Experimental characteristics for deformation properties of backfill mass. Mining of Mineral Deposits, 14(3), 119-127. https://doi.org/10.33271/mining14.03.119
2. Takhanov, D., Muratuly, B., Rashid, Z., & Kydrashov, A. (2021). Geomechanics substantiation of pillars development parameters in case of combined mining the contiguous steep ore bodies. Mining of Mineral Deposits, 15(1), 50-58. https://doi.org/10.33271/mining15.01.050
3. Ghazdali, O., Moustadraf, J., Tagma, T., Alabjah, B., & Amraoui, F. (2021). Study and evaluation of the stability of underground mining method used in shallow-dip vein deposits hosted in poor quality rock. Mining of Mineral Deposits, 15(3), 31-38. https://doi.org/10.33271/mining15.03.031
Point 16. In general, I must admit that a very good study was performed, and I will recommend your paper for publication after careful revision.

Round 2
Reviewer 3 Report
Dear authors,
I am more than satisfied with the corrections provided by you.
This study is an important contribution to sustainable mining.